# Natural Ventilation for Cooling Energy Saving: Typical Case of Public Building Design Optimization in Guangzhou, China

Menglong Zhang, Wenyang Han, Yufei He, Jianwu Xiong and Yin Zhang *

School of Architecture, Southwest Minzu University, Chengdu 610225, China
* Correspondence: cdzhangyin@163.com; Tel.: +86-134-8891-8589

**Featured Application: The architectural design, modeling, mechanical dynamic simulation, and corresponding analysis have been applied to a practical new construction program, a science museum located in Guangzhou, China.**

**Abstract:** Heating ventilation and air conditioning systems account for over one-third of building energy usage, especially for public buildings, due to large indoor heat sources and high ventilation and thermal comfort requirements compared to residential buildings. Natural ventilation shows high application potential in public buildings because of its highly efficient ventilation effect and energy-saving potential for indoor heat dissipation. In this paper, a building design is proposed for a science museum with atrium-centered natural ventilation consideration. The floor layout, building orientation, and internal structure are optimized to make full use of natural ventilation for space cooling under local climatic conditions. The natural ventilation model is established through computational fluid dynamics (CFD) for airflow evaluation under indoor and outdoor pressure differences. The preliminary results show that such an atrium-centered architectural design could facilitate an average air exchange rate over $2\,\text{h}^{-1}$ via the natural ventilation effect. Moreover, indoor thermal environment simulation results indicate that the exhaust air temperature can be about $5\,°\text{C}$ higher than the indoor air mean temperature during the daytime, resulting in about 41.2% air conditioning energy saving ratio due to the free cooling effect of natural ventilation. This work can provide guidance and references for natural ventilation optimization design in public buildings.

**Keywords:** public building; natural ventilation; wind pressure; energy saving; indoor environment

## 1. Introduction

The building construction sector is considered one of the largest consumers of natural resources and energy. Buildings consume 30–40% of all primary energy and natural resources over their lifespan (construction, operation, maintenance, and demolition) and account for 30% of the global emissions of greenhouse gases [1]. A significant proportion of the increase in energy use was due to the spread of HVAC installations in response to the growing demand for better thermal comfort within the built environment. In general, in developed countries, HVAC is the largest energy end-use, accounting for about half of the total energy consumption in buildings, especially non-domestic buildings [2]. Public building energy consumption constitutes a large proportion of total building energy use. The overall energy consumption per m$^2$ in public buildings is over two times higher than that in residential buildings in China [3]. Meanwhile, China faces constant growth in the amount of public building area and energy consumption [4]. The energy consumption intensity of public buildings has also increased rapidly. Heating ventilation and air conditioning (HVAC) systems account for nearly 50% of the energy consumption of public buildings [5]. In recent years, China has been undergoing fast urbanization, with the public building areas reaching 11.6 billion m$^2$ by 2020, and urbanization causes thermal elevation which increases household energy consumption through air conditioning to reduce human heat

stress [6,7]. In fact, in the building sector, HVAC systems represent between 40 and 60% of energy consumption in Europe and more than 50% in the United States [8]. Therefore, to reduce building energy, it is essential to reduce heating and cooling energy consumption. Public buildings mainly adopt concentrated air conditioning systems. For public buildings with certain thermal physical properties of the building envelope, there are two main factors affecting the cooling and heating load requirements [9]. One main impact factor is the outdoor climatic conditions (such as outdoor air temperature and humidity, and solar radiation intensity), which determine the basic amount of the cooling and heating load [10]. In particular, large-scale structures such as science and technology museums typically rely on air conditioning systems for indoor temperature control, a practice associated with increasing energy consumption. In alignment with national carbon reduction policies, there is a growing emphasis on harnessing the natural ventilation potential in regions characterized by hot summers and warm winters [11]. Studies have shown that occupants of naturally ventilated buildings are comfortable over a wider range of temperatures than occupants of buildings with centrally controlled HVAC systems [12]. Hence, many researchers have focused on natural ventilation system modeling, testing, and performance simulation of natural ventilation systems. Li and Chen [13] conducted a study on when and where natural ventilation cooling can be considered in the pre-design phase in China, and concluded that Kunming is the top priority for one-year mixed-ventilation air-conditioned buildings through a study of 100 cities. Zhang et al. [14] and González-Cruz et al. [15] combined the natural ventilation system with advanced air conditioning terminal devices such as floor radiation heating for an integrated system configuration investigation. Krusaa et al. [16] utilized building performance simulation software to analyze the effectiveness of natural ventilation in a local public building. The authors proposed a strategy that combines natural ventilation with air conditioning and verified that this strategy could reduce energy consumption by 13.1%. Ji et al. [17] conducted an analysis of the impact of air pollution and climate changes on the potential for natural ventilation in 74 Chinese cities from 2014 to 2019, identifying key factors limiting the utilization of natural ventilation. Maite Gil-Baez et al. [18] analyzed and compared the effectiveness of air renewal in two school buildings with a mechanical ventilation system compared to a natural ventilation system, and showed that by using a natural ventilation system, energy savings of 18–33% could be achieved while maintaining classroom comfort. Furthermore, Rosato et al. [19] and Prieto et al. [20] conducted a comprehensive summary of various natural ventilation methods and the current status of natural cooling methods, pointing out the research direction of achieving natural cooling through natural ventilation.

Considering the energy-saving potential of natural ventilation in the early stages of building design, especially in China, is currently the subject of more limited research. Kyosuke Hiyama et al. proposed a preliminary design methodology for naturally ventilated buildings utilizing a target air change rate [13].

The heavy dependence of sizable public structures on air conditioning systems for maintaining indoor thermal comfort leads to substantial energy consumption within these buildings. Consequently, mitigating the energy use of air conditioning systems in public structures has emerged as a pressing concern demanding immediate attention. However, although some scholars have considered the impact of natural ventilation potential on building energy efficiency in the early stages of building design, research based on a typical case of considering the cooling potential of natural ventilation in the early stages of building design is still insufficient.

Even though existing studies offer various concepts, methods, and ideas for building ventilation and air conditioning systems, with comprehensive consideration from experimental, technical, economic, and thermal perspectives, attention has been paid mainly to natural ventilation performance and operation from the viewpoint of the system designer. Designers of air conditioning systems pay relatively limited attention to air conditioning energy consumption, and reducing it is considered a challenging task. Therefore, the authors of this paper suggest that it is possible to approach this issue from the perspective

of architects. By incorporating considerations of building energy consumption in the early stages of architectural design, it becomes feasible to effectively reduce overall building energy consumption. This approach offers an effective reference and design strategy for achieving energy efficiency in buildings at the architectural design level. Consequently, this paper addresses several key questions: How can internal land use and space planning of buildings be determined from the perspective of building design as a way to reduce energy consumption? How can building design be optimized to leverage natural ventilation in local wind environments? What could be the pre-estimated energy-saving potentials of such naturally ventilated public buildings, compared to traditional reference design modes? This study presents tentative answers to these questions using the new construction project of a science museum in Guangzhou, China, as a typical illustrative example. The main body of the paper is structured as follows: (1) The natural ventilation dynamic models are established based on the local wind environment and climatic conditions. (2) The detailed architectural design and space planning are demonstrated and optimized with an atrium-centered floor layout to make full use of the free cooling effect through natural ventilation. (3) The energy saving potential for such a design is evaluated with dynamic building load simulation and annual energy consumption comparison. This work can provide a typical design reference and application prototype for new construction with natural ventilation considerations, especially for big public buildings with air conditioning energy efficiency improvement expectations.

## 2. Methods

### 2.1. Natural Ventilation Mechanism

Building ventilation is a key passive strategy for designing energy-efficient buildings and improving indoor air quality [21]. In the natural environment, the ventilation phenomenon in which a building relies on the airflow between the indoor and outdoor spaces to introduce fresh air and remove dirty air is usually called natural ventilation. Natural ventilation is usually referred to as the behavior of indoor personnel consciously opening building vents during the use of the building (especially during the transition season) to make full use of natural air to cool and dehumidify the building, thus reducing the indoor cooling and humidity loads of the building; air infiltration is usually defined as the physical phenomenon of air flowing from high-pressure to low-pressure areas through cracks, crevices, or non-deliberate openings in the enclosure structure during the operation phase of the building (e.g., air conditioning and heating seasons) when there is a difference in pressure between the indoor and outdoor spaces. This is the physical phenomenon of high-pressure to low-pressure flow [22]. Natural ventilation can be defined as a type of anticipatory ventilation, whereas air infiltration is a disorganized and unintended exchange of air. Based on their physics, both intended ventilation and unintended air infiltration are, in essence, natural ventilation. The driving force for natural ventilation is usually the wind pressure created by atmospheric movement on the building surface or the thermal pressure (which is essentially a difference in air density) caused by the difference in air temperature between the interior and exterior of the building [23]. Therefore, heat pressure and wind pressure are the two main factors affecting natural ventilation. In addition, another necessary condition for natural ventilation is the existence of cracks or openings in the building envelope for airflow. Ventilation caused by wind-driven forces is called wind-pressure ventilation, and ventilation caused by buoyancy-driven forces is called thermal-pressure ventilation.

Thermal pressures are caused by density differences due to differences in air temperatures between the interior and exterior of the building, and their effect on the building is known as the chimney effect. The root cause of the chimney effect in high-rise buildings is the temperature difference between indoor and outdoor air, which further leads to a density difference between indoor and outdoor air. This density difference results in a difference in the gravity of the indoor and outdoor air columns, which in turn results in a difference in the distribution of the static pressure gradient between the indoor and outdoor

air, resulting in a difference in air pressure. The greater the temperature difference between indoor and outdoor air when the building height is certain, or the higher the building is when the temperature difference between indoor and outdoor air is certain, the greater the heat pressure generated and the more significant the chimney effect caused.

When natural wind blows towards the building, due to the shielding of the building, the outdoor airflow will change direction around the building to produce a winding flow. The windward side of the building causes the airflow to be obstructed, resulting in the formation of a positive pressure zone, which is characterized by a decrease in dynamic pressure and an increase in static pressure. The top and sides of the windward side of the building force the air to produce a flow boundary layer on the surface of the building, which develops into a localized vortex, resulting in a reduction in static pressure on the top and back of the building, or even a negative pressure [23]. As a result of the atmospheric movement, the static pressure of the air around the building changes, and this pressure is often referred to as wind pressure. Wind pressure forces a differential pressure on both sides of an opening or cracks in the envelope, which manifests itself as a flow of air from a high-pressure area to a low-pressure area (Figure 1).

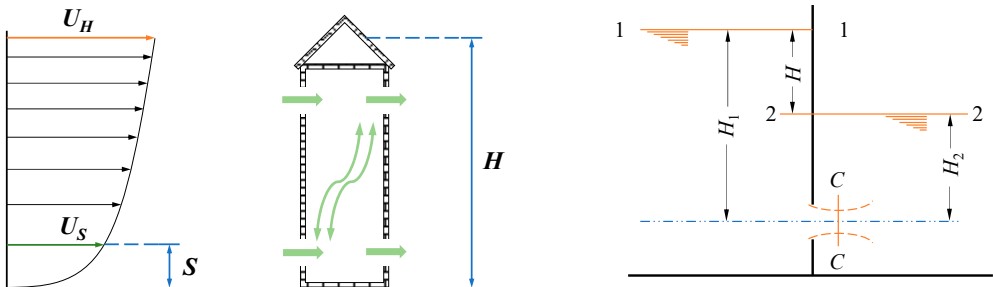

**Figure 1.** Schematic diagram of building natural ventilation driven by pressure difference.

## 2.2. Mathematical Modeling

The strength of wind pressure on a building surface is closely related to outdoor wind speed, wind direction, temperature, air density, and other parameters. However, in the actual environment, wind conditions are characterized by rapid changes and unpredictability, and the wind pressure on a building surface is strongly fluctuating and transient. Due to the complexity and variability of wind pressure, the accurate measurement of wind pressure on a building surface is a challenging task, especially when accurate transient wind pressure data are required. The airflow field around a building is related to the influencing factors such as the shape of the building, the angle between the building orientation and the wind direction, and the shading of the surrounding environment (e.g., buildings, structures, and vegetation), which will result in the wind pressure coefficient on the surface of the building being difficult to obtain directly from theoretical calculations. From the above analysis, it can be seen that both expected natural ventilation and unintended air infiltration are natural ventilation in nature [23]. According to the findings of fluid mechanics, the physical model of natural ventilation is the gas orifice flooding outflow model. As Figure 1 shows, according to the Bernoulli equation,

$$H_1 + \frac{P_1}{\rho g} + \frac{\alpha_1 v_1^2}{2g} = H_2 + \frac{P_2}{\rho g} + \frac{\alpha_2 v_2^2}{2g} + \zeta_1 \frac{v_c^2}{2g} + \zeta_2 \frac{v_c^2}{2g}, \tag{1}$$

Then, the airflow velocity caused by natural ventilation can be expressed by

$$v_c = \frac{1}{\sqrt{\zeta_1 + \zeta_2}} \cdot \sqrt{2gH_0}, \tag{2}$$

where $\zeta_1$ is the local resistance coefficient of the liquid through the orifice and $\zeta_2$ is the local resistance coefficient of the liquid that suddenly expands after the contraction of the

section ($\zeta_2 = 1$ because the 2–2 section is much larger than the C–C section). Air outflow is generally submerged outflow, due to the small gas capacity, which can be ignored in the total head difference before and after the orifice of the location of the head term (bit pressure). Furthermore, when there is a pressure difference between the two sides of the geometrically large openings in the building envelope, the natural ventilation airflow pattern is turbulent flow. Thus, after taking the building opening area into consideration, the ventilation airflow rate is

$$Q_V = \mu A \sqrt{\frac{2\Delta P_0}{\rho}}, \tag{3}$$

where $\rho$ and $\mu$ are density (kg/m$^3$) and opening airflow coefficient, respectively. Currently, the commonly used methods for determining the wind pressure coefficient mainly include field measurements, wind tunnel experiments, computational fluid dynamics numerical simulation (CFD), or semi-empirical models [24].

According to the Navier–Stokes fluid mechanic theory, partial differential equations can be deduced for incompressible fluid and flows. Generally speaking, natural ventilation generates turbulent airflow in big public buildings, which makes it difficult to solve Navier–Stokes equations because of the greatly varying mixing-length scales for any turbulent airflow [25]. In this paper, the $k$–$\varepsilon$ turbulence model is used to govern the Reynolds-averaged Navier–Stokes (RANS) formulas, combined with practical computational fluid dynamics (CFD) applications. The mass continuity equation of incompressible fluid is:

$$\frac{d\rho}{dt} + \nabla \cdot (\rho V) = 0, \tag{4}$$

where $V$ is the flow velocity vector (m/s). The airflow density remained unchanged:

$$\nabla \cdot (\rho V) = \sum_i \frac{\partial \rho V_i)}{\partial x_i} = -\frac{d\rho}{dt} = 0. \tag{5}$$

Then, according to Newton's second law of motion,

$$\frac{d(\rho V)}{dt} = F + \nabla \cdot P, \tag{6}$$

where $F$ and $P$ represent body force and mass force, respectively. Then, integrated with viscous stress, the momentum equation can be expressed by

$$\frac{\partial(\rho V)}{\partial t} + V_j \frac{\partial(\rho V_i)}{\partial x_j} = F - \frac{\partial P}{\partial x_i} + \mu \nabla^2 V, \tag{7}$$

where $\mu$ is the kinematic viscosity (kg/m$^2$s) and $P$ is the static pressure (Pa). For the air energy transfer and conversion,

$$\frac{\partial \rho H}{\partial t} + \frac{\partial}{\partial x_i}(V_i \rho H) = \frac{\partial}{\partial x_j}\left(\frac{\lambda}{c_p} \frac{\partial H}{\partial x_j}\right) + S, \tag{8}$$

where $H$, $\lambda$, and $c_p$ represent the enthalpy (kJ/kg), thermal conductivity (W/mK), and specific heat (kJ/kgK), respectively, and $S$ represents the heat source (kW). According to the Boussinesq eddy viscosity hypothesis, the turbulent model using the $k$–$\varepsilon$ model changes into

$$\rho \frac{\partial K}{\partial t} + \rho V_i \frac{\partial K}{\partial x_i} = \frac{\partial}{\partial x_i}\left[\left(\mu + \frac{\mu_t}{\sigma_k}\right)\frac{\partial K}{\partial x_i}\right] + \mu_t \frac{\partial V_i}{\partial x_j} \nabla \cdot V - C_a \rho \frac{k^{\frac{3}{2}}}{l}, \tag{9}$$

where $C_a$ is a constant, and $\mu_t$ and $\sigma_k$ represent the kinematic viscosity for turbulent flow and Prandtl number of turbulence kinetic energy, respectively. The heat dissipation in the turbulent airflow can be depicted by

$$\rho \frac{\partial \varepsilon}{\partial t} = \frac{\partial}{\partial x_i}[(\mu + \frac{\mu_t}{\sigma_k})\frac{\partial \varepsilon}{\partial x_i}] + C_b \frac{\varepsilon}{K}(G_k + C_c) - C_d \rho \frac{\varepsilon^2}{K}. \tag{10}$$

The above governing equations can be recast in a conservative form, and then solved through the finite volume method, combined with the semi-implicit method for pressure-linked equation-consistent algorithm [26].

### 2.3. Air Exchange Rate

This project adopts the multi-area network method to calculate the number of indoor air changes in the building. In the multi-area network method, the indoor rooms are divided into different ventilation and air changes in different areas, with the wind pressure of windows and doors as the boundary conditions; the different areas are connected through the connected windows and doors for the transmission of data, and ultimately to obtain the number of air changes in each room.

The calculation of the number of air changes in the room is derived from the calculation of the air quality flow in the ventilation path, and the air quality flow based on the multi-area network method is calculated as follows:

$$Q = C_d A \sqrt{\frac{2\Delta P}{\rho}} \tag{11}$$

where $Q$ represents the room volume flow ($m^2/s$); $P$ represents the difference in wind pressure between the doors and windows of adjacent rooms; and $C$ represents the flow coefficient, which, for large building openings, is 0.5, for narrow openings, 0.65, and for the project calculations, 0.6; $A$ represents the area of the opening ($m^2$), $\rho$ represents the density of air ($kg/m^2$).

After obtaining the volume flow rate $Q$ of a room by the above method, the calculation of the number of air changes in the room can be carried out:

$$Acr = \frac{Q \times 3600}{V} \tag{12}$$

where $Q$ denotes the room volume flow rate ($m^3/s$); $Acr$ denotes the number of air changes (times/h); and $V$ denotes the room volume ($m^3$).

In the following section, an atrium-based natural ventilation system in a newly designed science museum in Guangzhou is taken as the illustrative example to investigate the free cooling potential of pressure-difference-driven ventilation under local climatic conditions for building thermal design optimization.

### 2.4. Simulation and Solution

CFD analysis was performed using a polyhedral mesh grid generated through ANSYS Meshing (Ansys-Fluent v22.2). Considering the change in outdoor wind direction, the inlet and outlet of wind speed in the geometric model will be changed, and the power shadow area has a large impact on the building surroundings, so the size of the calculation area has a significant impact on the accuracy of the numerical calculation results. If the computational domain is too small it will cause problems of flow field distortion and insufficient flow development, and if the computational domain is too large it will cause unnecessary waste of computational resources [23]. Regarding the setting of the computational domain, the COST guideline stipulates that for the simulation of a single building, the distance between the two sides of the building and the two sides of the computational domain should be more than 5 times the height of the building, the distance between the top of the building

and the top boundary of the computational domain should be more than 5 times the height of the building, and the distance between the boundary of the air outlets and the building should be at least 15 times the height of the building. Based on the above considerations, the dimensions of the computational domain for the simulation in this study are set as shown in Figure 2. Taking the base model as an example, the distance of the wind speed inlet from the building is usually 5 H (H is the building height), the distance of the outlet from the building is 10 H, and the boundaries of the left and right calculation domains are usually 5 H from the building. Before conducting outdoor wind field calculations, it is essential to determine the size of the computational domain involved, termed the computational domain in fluid mechanics, which typically constitutes a rectangular or cubic space encompassing a cluster of buildings.

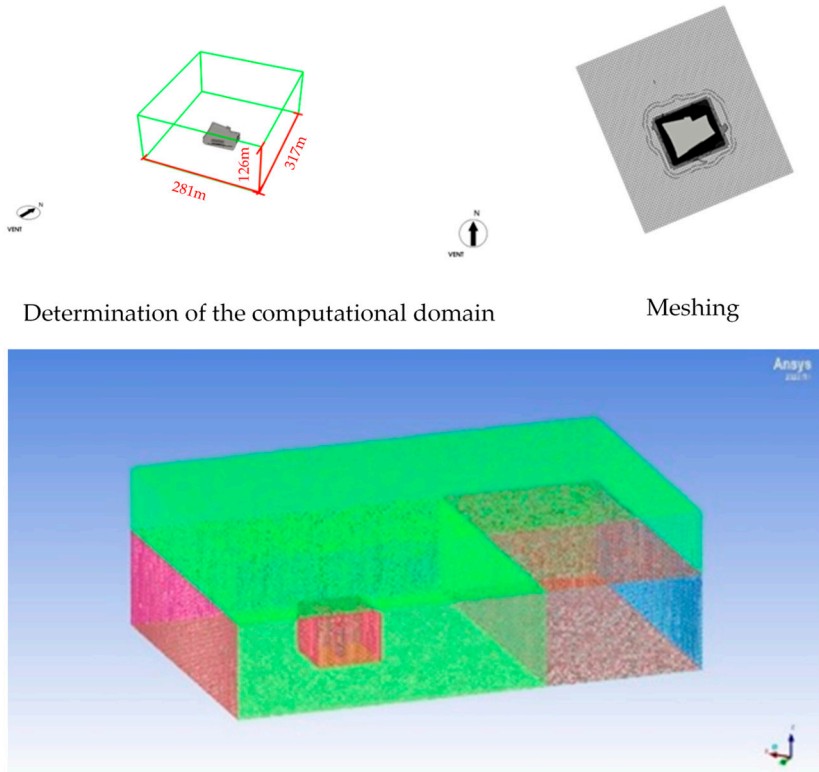

Determination of the computational domain    Meshing

**Figure 2.** Schematic diagram of CFD simulation and meshing (colours difference denoting computational domains for building surrounding zones and city wind environment respectively).

Figure 2 displays a cross-sectional diagram of the wind field computational domain, the grid generation, and the allocation of boundary conditions for each region. The dimensions of the wind field are 317 m in length, 281 m in width, and 126 m in height. The total grid count amounts to 1,687,448.

Figure 3 illustrates the research workflow of this study. Initially, the objective was to explore the potential of natural ventilation in reducing temperatures and conserving energy, coupled with geographical and climatic data for preliminary architectural design. Subsequently, from an architectural perspective, energy-saving considerations were incorporated, encompassing the preliminary design of ventilation corridors, architectural self-shading, and the augmentation of ventilation. Thirdly, an architectural model was established using simulation software, followed by the configuration of various parameters for simulation. Finally, the simulation results validated the substantial energy-saving effects of the architectural energy-saving methods considered during the initial design phase.

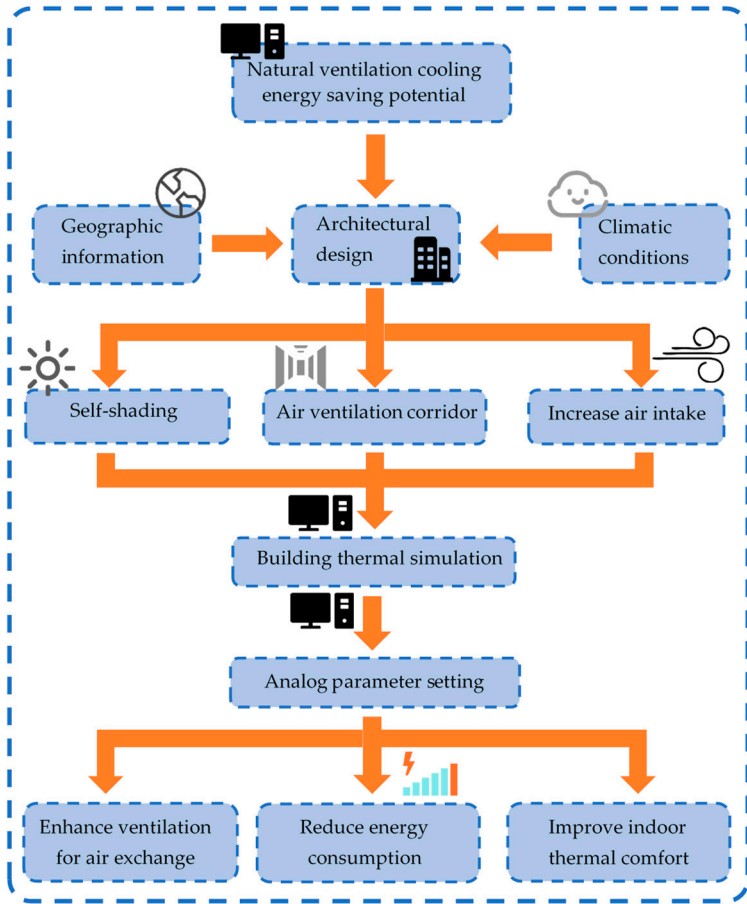

**Figure 3.** Research flow chart.

## 3. Geographical Information

Guangzhou City is situated in the southern region of mainland China, in the central and southern parts of Guangdong Province, at the northern edge of the Pearl River Delta. It is the confluence point for the Beijiang (North River), Xijiang (West River), and Dongjiang (East River) as they flow into the South China Sea. Geographically, the city is positioned between approximately 112°57′ to 114°3′ E longitude and 22°26′ to 23°56′ N latitude (Figure 4).

This paper focuses on the hottest region with hot summers and warm winters among China's five climate zones, where cooling consumption is an important component of building operation consumption [27,28]. So, Guangzhou was selected as the illustrative example with the following main climate parameters: the annual average temperature is 22.5 °C, the average temperature in the hottest month is 32.9 °C, the annual maximum temperature is 36.1 °C, and the annual total solar radiation is 1072.1 kWh/year. Figure 5 gives the key climatic parameter variations throughout a typical year. China divides building climate zones into five main climate zones and 20 climate subzones. The main division index of building climate zones is the average monthly temperature in January and July. The names of thermal zones are cold region, hot summer and cold winter region, hot summer and warm winter region, and mild region. The case study building is located in Guangzhou City and belongs to the hot summer and warm winter region, that is, the average temperature in January is greater than 10 °C, and the average temperature in July is 25–29 °C. Guangzhou falls within a subtropical climate zone, characterized by distinct seasons and abundant precipitation. Summer, the rainy season, is hot and humid, influenced by tropical monsoons. Winter is warm, dry, relatively mild, and experiences minimal rainfall, shaped by the influence of northern monsoon winds. Spring and autumn boast moderate temperatures and lower humidity levels.

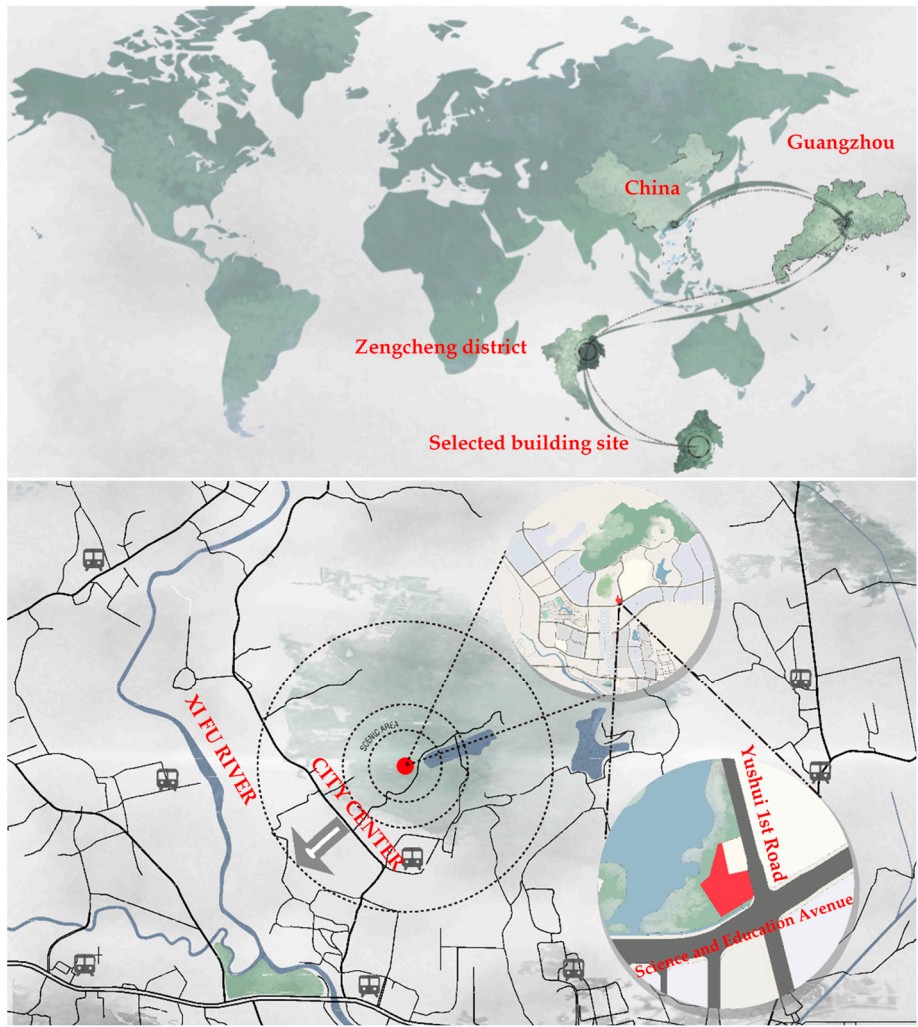

**Figure 4.** Location map of the case study city, Guangzhou, in southern China (drawn by authors).

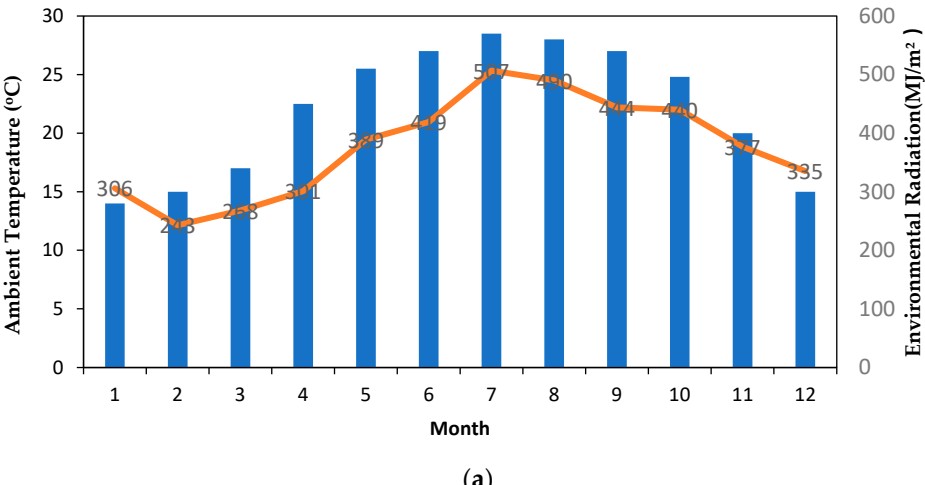

(**a**)

**Figure 5.** *Cont.*

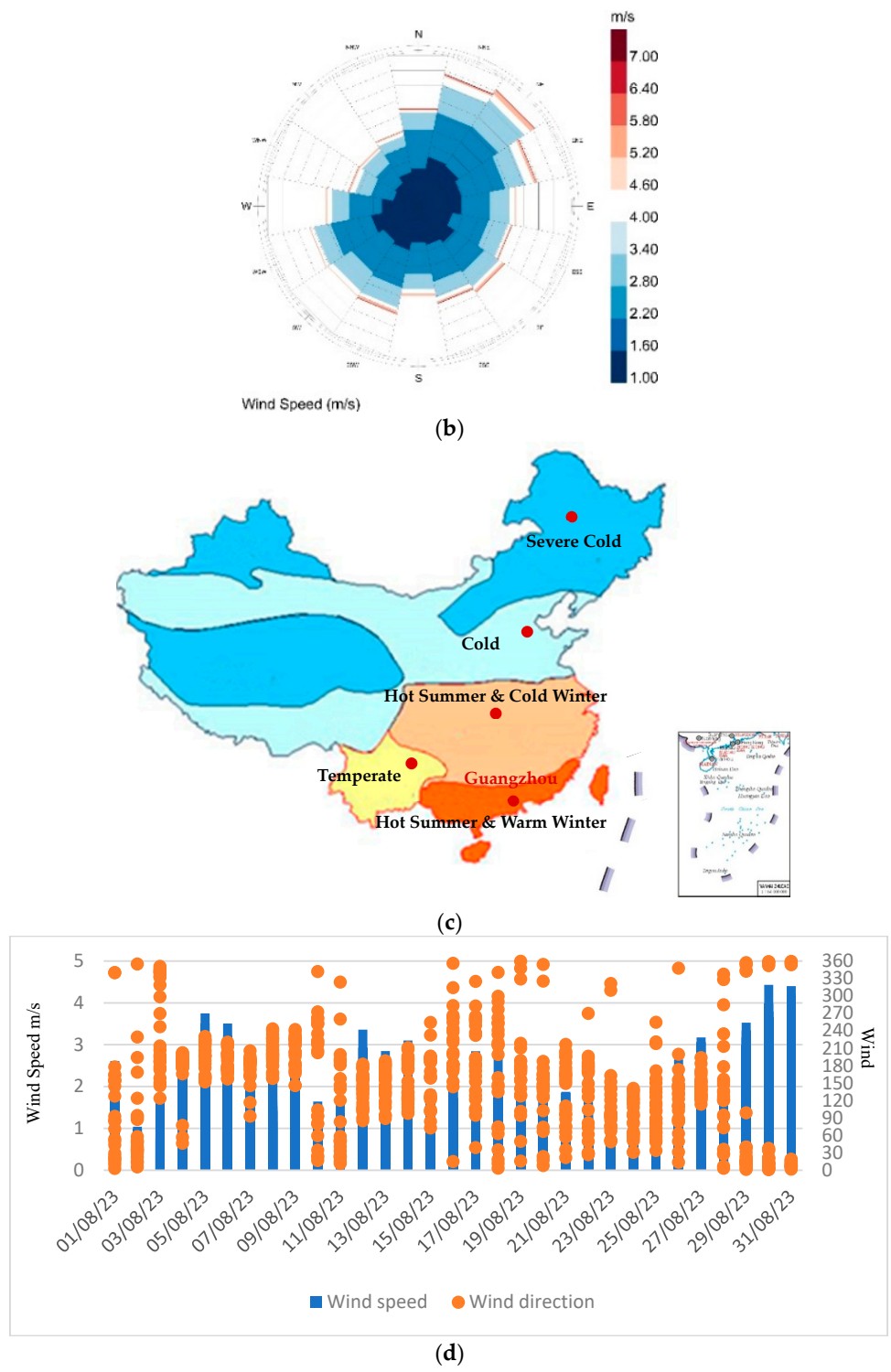

**Figure 5.** Key climatic parameters in Guangzhou (hot summer and warm winter climatic zone). (**a**) Average monthly temperature and solar radiation (Monthly average temperature in blue, monthly ambient radiation in orange). (**b**) Wind rose diagram (The color difference represents the wind speed in each wind direction zone). (**c**) Map of China's 5 climate zones (The color difference represents different climate zones in China) [29]. (**d**) Information on the wind environment at the site during a typical summer month (Blue for wind speed, orange for wind direction).

Guangzhou experiences its highest annual temperatures in August, reaching 37 °C. During the summer, temperature fluctuations are relatively minor. The lowest temperatures

occur in January and February, dropping to 6 °C, and during the winter, temperature variations are more significant. The monthly normal direct radiation remains around 250 MJ/m$^2$ with relatively minor annual fluctuations. The monthly total horizontal radiation exhibits a trend of transitioning from low to high and then decreasing from January to December, reaching its peak in August at approximately 530 MJ/m$^2$, and reaching its lowest point in February at approximately 260 MJ/m$^2$. The wind rose diagram indicates the predominant wind directions are from the northeast and southwest in Guangzhou. Specifically, the average wind speed for the northeast wind can reach a maximum of 6.40 m per second, while the southwest wind reaches a maximum average speed of 5.80 m per second. It is evident that the average wind speed for the northeast wind is significantly higher than that for the southwest wind.

In Figure 5d, the wind speed and direction around the building site during typical summer months are depicted. In Figure 5d, wind speed recordings were taken at one-hour intervals each day, and the maximum daily wind speed was used as the value for Figure 5d. And according to the statistics, the lowest recorded wind speed for the month was 0.09 m/s, while the highest was 4.4 m/s. The scatter plot in the figure indicates that the predominant wind direction during this period is concentrated between 90° and 180°, primarily coming from the southeast.

## 4. Results

### 4.1. Building Design Optimization

The chosen study case here is a typical science museum, with a total building area of 11,447 m$^2$, a volume of 55,726.81 m$^3$, and shape coefficient of 0.30 (Table 1). The technology museum implemented several strategies to enhance the cooling and energy-saving potential through natural ventilation. These included the use of ventilation corridors, structural shading elements integrated into the façade from top to bottom, and an expanded main entrance to augment natural airflow. Serving as a representative case study, it demonstrates the application of these tactics to encourage natural ventilation in buildings and effectively reduce energy consumption.

**Table 1.** Building thermal physical properties.

| Parameters | Value | Unit |
| --- | --- | --- |
| total area | 19,012.41 | m$^2$ |
| external envelope area | 7771.2 | m$^2$ |
| building volume | 55,726.81 | m$^3$ |
| shape coefficient | 0.30 | |
| ratio of window to wall | 0.34 | |
| window opening area (south) | 265.83 | m$^2$ |
| heat transfer coefficient of external wall | 1.07 | W/m$^2$·K |
| heat transfer coefficient of external window | 3.5 | W/m$^2$·K |
| solar radiation absorption coefficient | 0.08 | |
| penetration coefficient of solar radiation through the glazing | 0.8 | |
| shading coefficient | 1 | |

As shown in Figure 6 the building's south façade gradually extends outward from the bottom to the top, achieving self-shading for the structure. This design embodies the concept of integrating gray spaces, shading, and epidermal ventilation. The outward expansion of the building's exterior structure serves the following key functions: Firstly, it provides a gray space for the building's main entrance, serving not only as a shading canopy structurally but also as a brief stopover and gathering space for people, further enhancing the functionality and pleasantness of the building's entrance. Secondly, the extension of the structure outward increases the inflow volume of the ventilation corridor, facilitating the

harnessing of the cooling potential of natural ventilation. This plays a vital role in improving the interior comfort of the science museum and reducing its energy consumption.

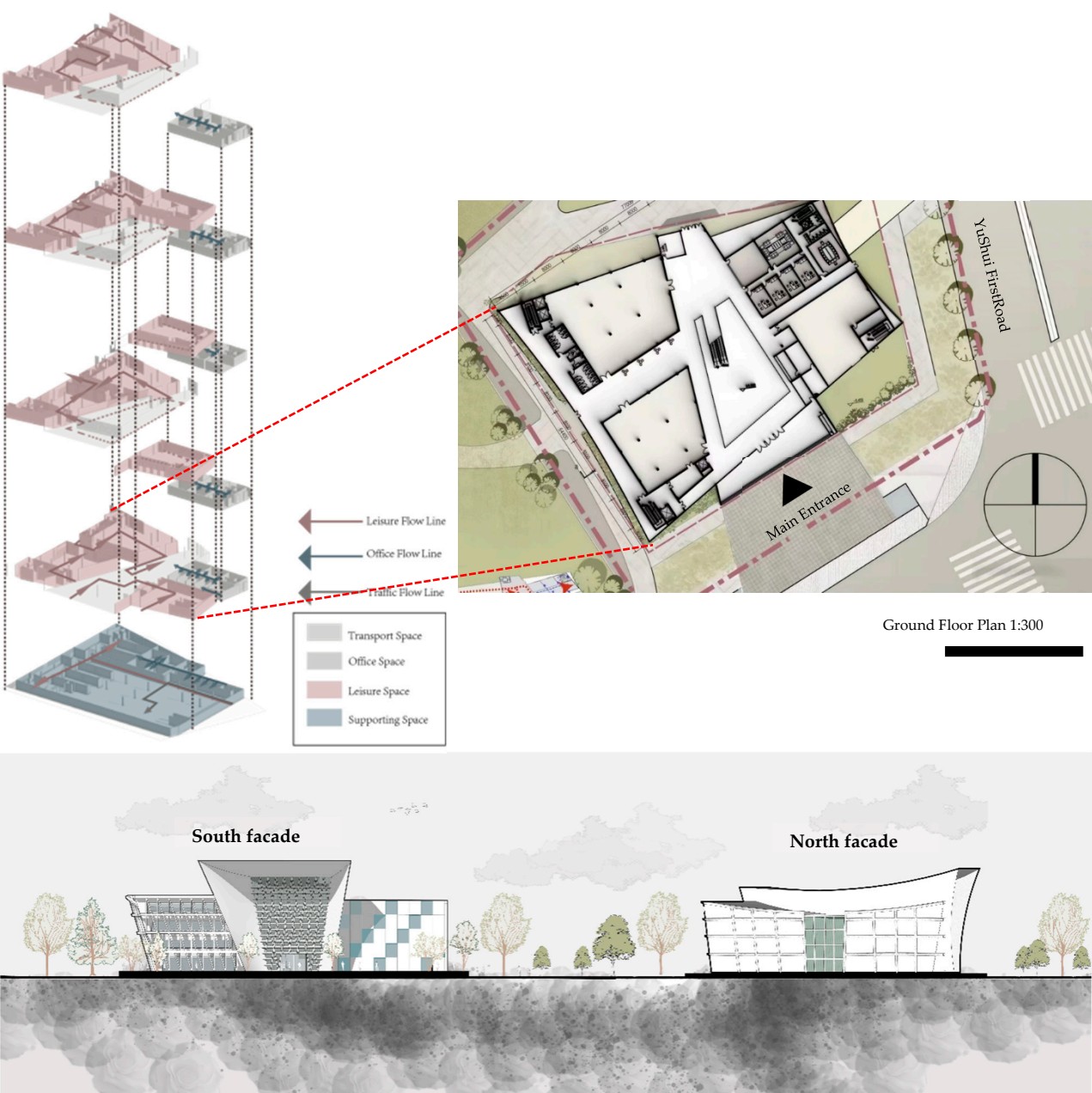

**Figure 6.** Architectural design details with floor layouts and external façades (drawn by authors).

Figure 6 depicts the ground floor plan of the science museum, where the primary functions of this level are categorized into exhibition areas and office spaces. Notably, the ground floor incorporates a gradually tapering north–south-oriented ventilation corridor. This ventilation corridor serves a dual purpose: firstly, it functions as a spatial divider within the architectural layout; secondly, it effectively enhances the natural ventilation capacity of the building, thereby facilitating the exploitation of the cooling potential inherent in natural ventilation. This architectural feature is strategically designed to improve the overall thermal performance of the building by optimizing natural airflow and cooling mechanisms.

Figure 7 illustrates the main passive building thermal design optimization approaches including natural ventilation, solar radiation absorption, smart shading and rain collection, etc. This pathway initiates from the main entrance on the southern façade of the building, and traverses through a ventilation corridor, driven by wind pressure, to fulfill the lateral ventilation needs of the structure. Simultaneously, natural air enters the interior through the central atrium of the building and influenced by thermal pressure, meets the vertical ventilation requirements of the indoor space. Notably, the glass surface at the top of the building's atrium serves as a rainwater collection system, contributing to the supplementary cooling of the interior. Additionally, the architectural extensions extending outward enhance the spatial capacity of the main entrance area, thereby increasing the effectiveness of natural ventilation.

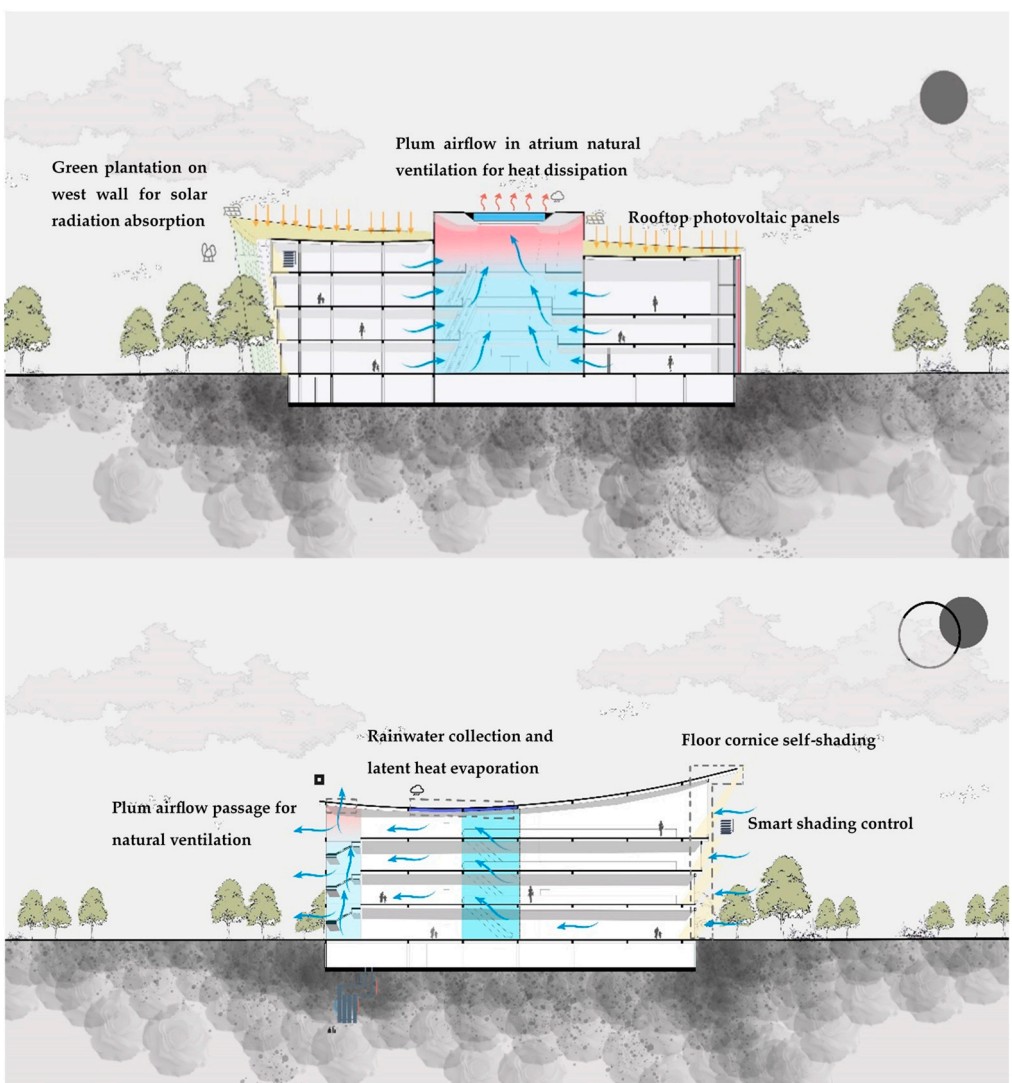

**Figure 7.** Schematic design mechanisms for passive thermal optimization (drawn by authors) (Color differences represent different bioclimatic technologies).

### 4.2. Natural Ventilation Simulation

The main focus of this study is investigating the cooling potential of natural ventilation and its energy-saving effects. Given that Guangzhou experiences a hot summer and warm winter climate, with monthly average temperatures consistently exceeding 10 °C, particularly reaching 28 °C in the summer, the proportion of air conditioning cooling consumption in total building energy consumption is significant. Therefore, simulations are set under summer conditions, with the external wind direction at that time being southeast.

Given that the primary focus of this study is on the isolated impact of natural ventilation on cooling and energy conservation, despite the utilization of various bioclimatic techniques in the building, they do not directly affect natural ventilation. Therefore, the energy simulation did not account for the influence of bioclimatic techniques on energy consumption.

Figure 8a represents a simulation of the outdoor wind environment around the building. Outdoor wind speeds are indicated in ascending order from blue to red. As observed in Figure 8a, when the wind is blowing from the southeast, no areas within the human activity region are marked with wind speeds less than 0.2 m/s, thus concluding that there are no windless zones within the building.

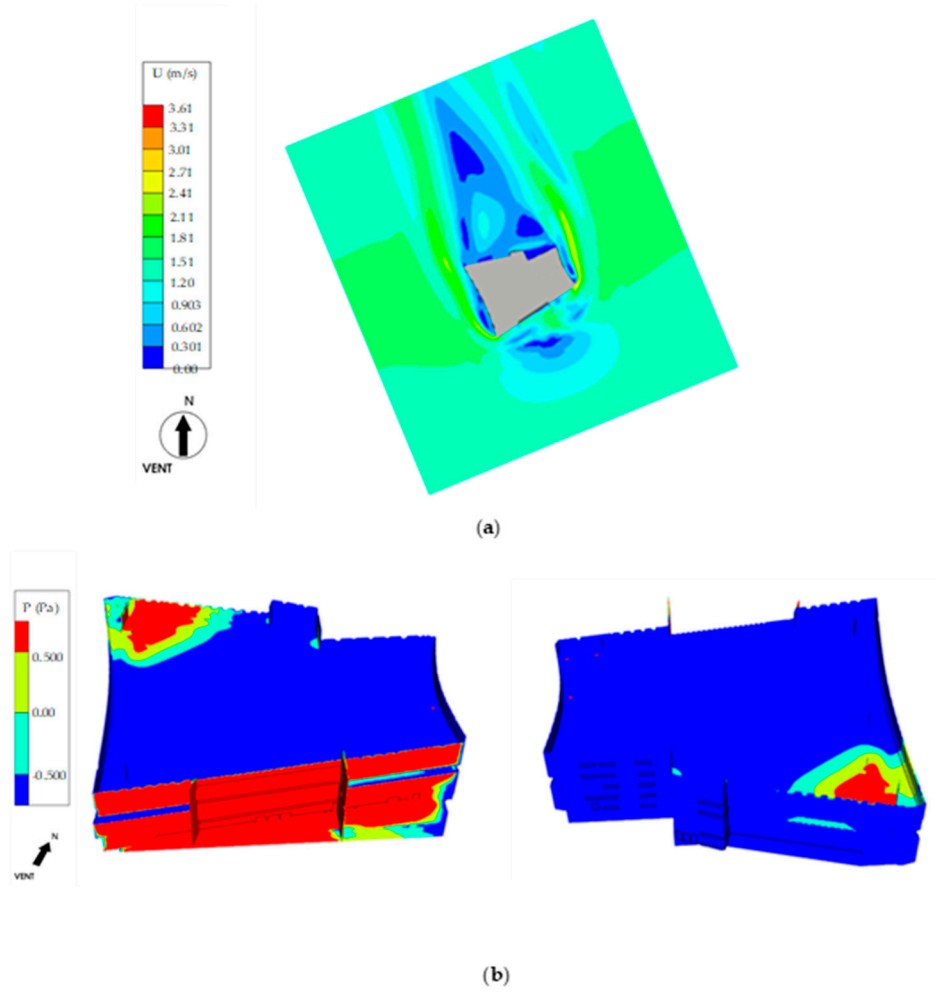

**Figure 8.** Outdoor wind environment chart. (**a**) outdoor wind speed. (**b**) building external wall surface air pressure.

Only when the absolute wind pressure on the exterior surfaces of the windows is sufficiently high can a good effect of window-based ventilation be ensured, leading to the creation of a favorable indoor airflow environment. Figure 8b displays the wind pressure distribution on the external window surfaces of the windward side of the building in summer conditions. This graph clearly conveys the wind pressure distribution on the windward side of the building. From Figure 8b, combined with the values in the illustration, it can be clearly seen that the wind pressure on the surface of the external window is less than 0.5 Pa in the external window area.

Figure 8b depicts the wind pressure distribution on the exterior window surfaces of the leeward side of the building in summer conditions, clearly presenting the wind pressure distribution pattern on the leeward side of the building. From Figure 8, combined with the values in the illustration, it can be clearly seen that the wind pressure on the surface of the

external window is less than 0.5 Pa in the external window area. The building is equipped with a total of 102 operable external windows, with 92 of them being affected by an outdoor wind pressure differential exceeding 0.5 Pa. Consequently, the window-based ventilation in this building exhibits excellent performance, contributing to improved indoor airflow quality. This provides a solid foundation for harnessing the potential of natural cooling through indoor ventilation.

Figure 9 shows a schematic diagram of the indoor wind environment. Figure 9a,b are the indoor wind speed map and indoor wind speed vector map of the building, respectively. Combined with the legend, it can be seen that most of the wind speeds in the ventilation corridor are greater than 0.5 m/s, which proves that the ventilation corridor increases the good effect on the amount of natural ventilation. Figure 9c is the natural wind flow line map of the interior of the building, which clearly reflects the excellent ventilation effect of the ventilation corridor. Figure 9d shows the addition of east–west windows to the original building, which clearly shows that the increase in the indoor east–west wind is larger, affecting the ventilation effect of the ventilation corridor. Figure 9b,c are compared, and combined with the outdoor southeast wind direction, it can be concluded that the practice of catering to the outdoor wind direction of the building's internal ventilation corridor effectively enhances the indoor ventilation volume and controls the direction of the natural wind into the room well.

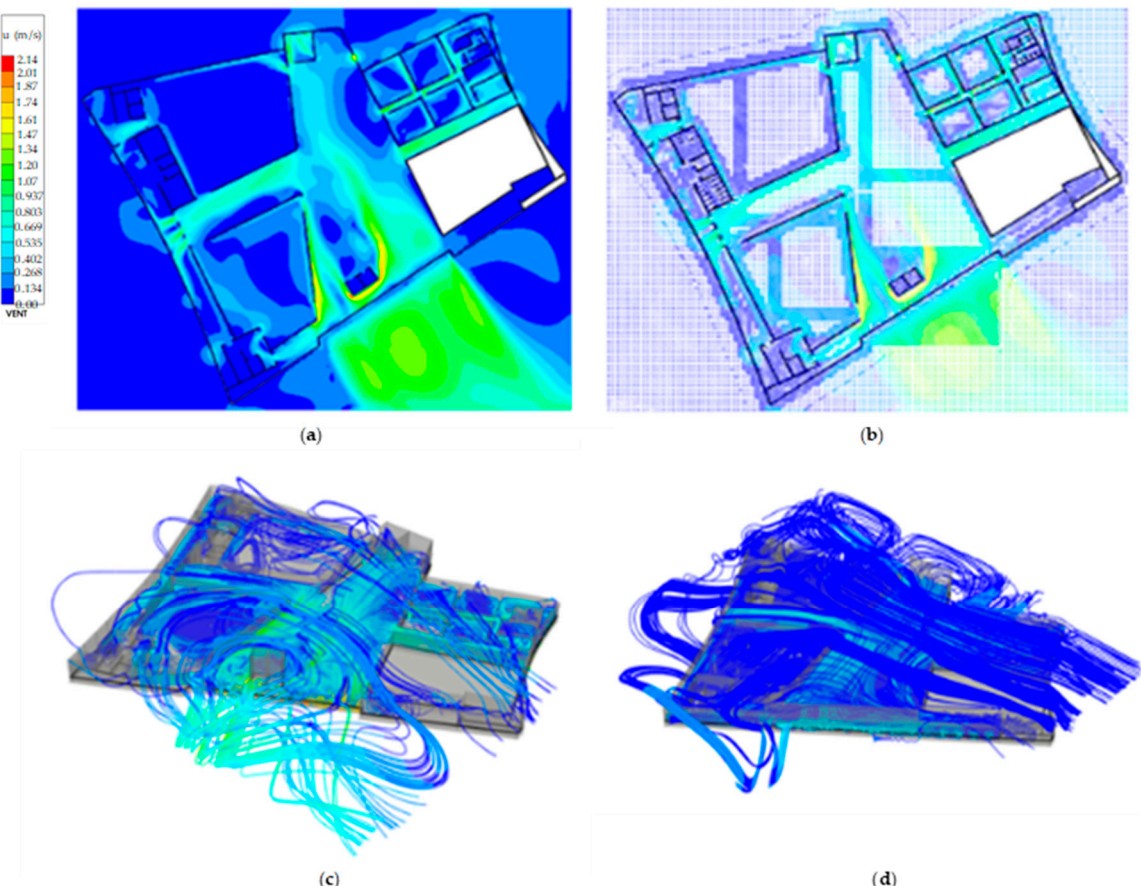

**Figure 9.** Indoor airflow simulation results under natural ventilation. (**a**) indoor air velocity. (**b**) indoor wind speed vector. (**c**) indoor airflow streamline. (**d**) Indoor airflow streamline (for comparison).

## 5. Discussion

The outdoor weather conditions of the office building are based on conditions in Guangzhou, China, which is in a hot summer and cold winter zone. The characteristic temperature method (CTM) is utilized for the research. Based on the building energy gene

theory [30], the dynamic load and energy consumption of buildings are simulated by CTM, and the relationship between load or energy consumption and various other factors can be revealed. According to CTM, if solar radiation gain is considered, the indoor characteristic temperature can be expressed by

$$T_{\text{in}} = \frac{\sum K_i F_i T_{si} + \sum F_i I \left( \eta_i + \frac{\alpha_i}{\alpha_o} \rho_G \right) C_i \mp Q_{\text{AC}}}{\sum K_i F_i} \tag{13}$$

where $F_i$ is the building envelope area, m$^2$; $T_{si}$ is the equivalent solar-air temperature, °C; $K_i$ represents heat transfer coefficient, W/(m$^2 \cdot$ K); $I$ represents solar radiation, W/m$^2$; $\eta_i$ is the transmittance; $\rho_G$ is the absorption ratio; and $C_i$ is the shading coefficient. $Q_{AC}$ is the cooling or heating capacity provided by the air conditioner to maintain the indoor setting temperature within the thermal comfort zone [10]. On the other hand, if not considering solar radiation, $I$ is equal to zero and $T_{si}$ is approximately the outdoor temperature $T_{out}$, °C. So, the expression is changed into

$$T_{\text{in}} = \frac{\sum K_i F_i T_{air} \mp Q_{\text{AC}}}{\sum K_i F_i} \tag{14}$$

This CTM approach for indoor temperature prediction and building load simulation has been validated in previous studies and verified by commercial building simulation software including DOE-2 (v 2.2, U.S. Department of Energy), EnergyPlus (v 23.2.0, U.S. Department of Energy) and DeST (v DeST-h, Tsinghua University, China) [11,19,31,32]. For the other working conditions of building simulation, such as indoor air exchange rate (ACH) through a mechanical ventilation system, internal heat gains from indoor lighting, occupancy and devices, etc., key parameters are obtained based on the benchmark values according to the U.S. Department of Energy (DOE) Reference Building and ASHRAE AS (2013) Standard of thermal environmental conditions for human occupancy (building's daily office hours are 9:00–17:00) [33–35]. For the indoor thermal comfort evaluation, the integrated uncomfortable degree $I_{\text{year}}$ is used as the index, which is defined as:

$$I_{\text{year}} = \int_{\substack{\text{year} \\ T_L > T_{\text{in}}}} (T_L - T_{\text{in}}) d\tau + \int_{\substack{\text{year} \\ T_{\text{in}} > T_H}} (T_{\text{in}} - T_H) d\tau \tag{15}$$

where $T_{\text{H}}$ (26 °C) and $T_{\text{L}}$ (18 °C) are the upper and lower indoor temperature limits of the thermal comfort zone, respectively. This evaluation index depicts the integrated degree of deviation of the indoor temperature from the thermal comfort zone [10,11].

Based on the aforementioned building thermal performance modeling, the average indoor air temperature variations can be obtained with comparison under natural ventilation optimization (Figure 10). According to the thermal comfort demands, the upper temperature limit is set at 26 °C and the lower temperature limit at 18 °C. When indoor temperatures fall within this range, individuals experience a state of warmth and comfort, without feeling excessively cold or hot. The blue line represents the indoor temperature variations under natural ventilation conditions. Based on annual hour statistics, indoor temperatures remain within the thermal comfort range (19–26 °C) for a total of 4760 h, occurring primarily in the time intervals of 0–3000 h and 7000–8760 h. Conversely, the green line illustrates indoor temperature fluctuations in the absence of natural ventilation. Similarly, based on annual hour statistics, indoor temperatures persist within the thermal comfort zone for a total of 3100 h, mainly falling within the time intervals of 1000–3000 h and 6700–7800 h. By comparing the durations of thermal comfort under these two conditions, it becomes evident that under natural ventilation, indoor temperatures remain within the thermal comfort zone for an additional 1660 h compared to conditions without natural ventilation, indicating a significant improvement in indoor thermal comfort. Furthermore, as observed in Figure 10, under natural ventilation conditions, the indoor minimum tem-

perature is relatively higher compared to conditions without natural ventilation, with a maximum temperature difference of 10 degrees.

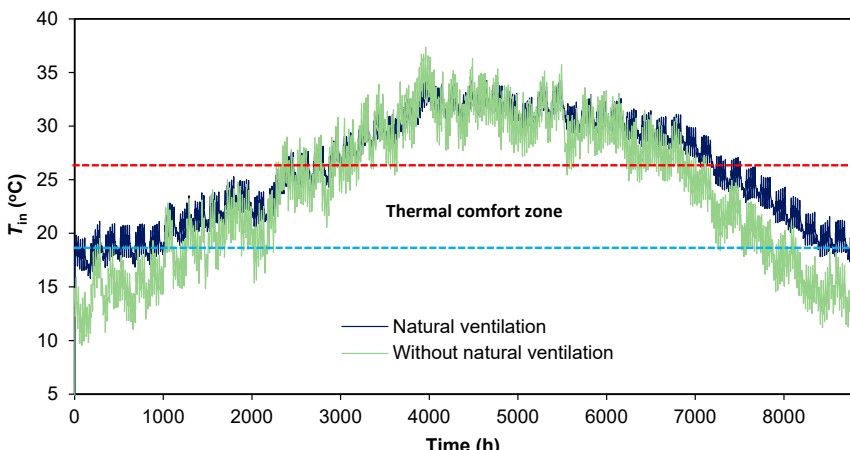

**Figure 10.** Indoor air temperature variations with thermal comfort comparison (red and blue lines represent the upper and lower temperature values for indoor thermal comfort zone).

Figure 11 compares the energy consumption of the designed building with that of the baseline building. In the simulated science museum building, the energy consumption primarily includes cooling, heating, lighting, and domestic hot water. Among these, lighting energy consumption constitutes 68% of the total building energy consumption, making it the highest among all energy consumption categories. Heating energy consumption has the lowest proportion, at only 2%, a difference attributed to the geographical location of the simulated building. Notably, in the simulated building located in Guangzhou, particular attention should be given to cooling energy consumption, which accounts for 29% of the total energy consumption.

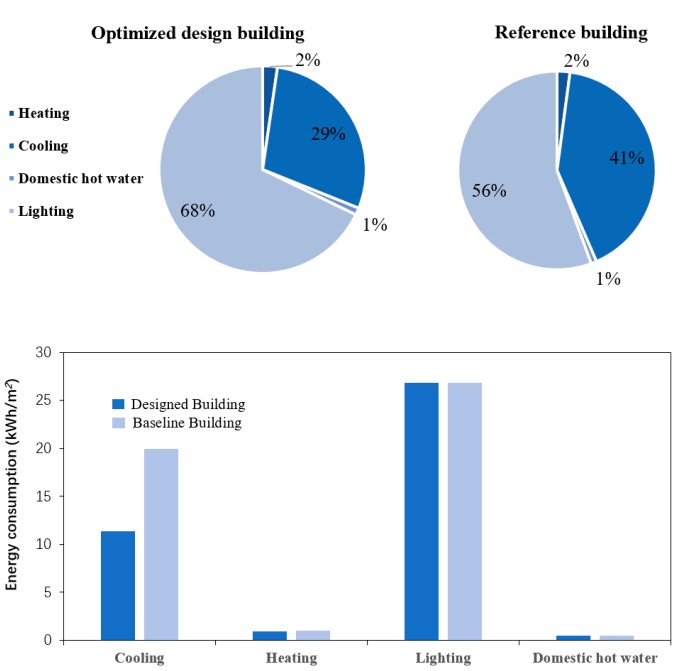

**Figure 11.** Building energy consumption proportions and saving potential evaluation.

In the energy composition of the baseline building, lighting also has the highest share of energy consumption in the baseline building, accounting for 56% of the total energy

consumption. Heating accounts for 2% and domestic hot water is the lowest at 1% as in the simulated building.

Since natural ventilation mainly affects air conditioning energy consumption in terms of energy saving, the design building reduces cooling energy consumption by 8.54 kWh/m$^2$ and heating energy consumption by 0.1 kWh/m$^2$ compared to the baseline building. Since the case is located in Guangzhou, China, a hot-summer and warm-winter zone, the increase in heating energy consumption is small, even though the temperature is not in the comfort temperature range in winter, the actual temperature is not much different from the minimum comfort temperature, and the heating demand is not large. The cooling energy consumption and total energy consumption of the design building are 11.37 kWh/m$^2$ and 39.55 kWh/m$^2$, respectively, and the cooling energy consumption accounts for 29% of the total energy consumption. Meanwhile, the cooling energy consumption and total energy consumption of the baseline building are 19.91 kWh/m$^2$ and 48.19 kWh/m$^2$, respectively, and the cooling energy consumption accounts for 41% of the total energy consumption. It can be seen that the proportion of refrigeration energy to total energy consumption in the design building is 12% lower than that in the baseline building. This disparity indicates that the simulated science museum building, through architectural design measures such as increasing ventilation corridors and enhancing natural ventilation, fully exploits the natural cooling potential, resulting in a significant 42.9% reduction in cooling energy consumption, thus achieving notable energy savings.

In comparison to the baseline building, the designed structure's air conditioning consumption stands at 12.3 kWh/m$^2$, while the baseline building's consumption reaches 20.94 kWh/m$^2$. This denotes a 41.2% reduction in air conditioning consumption for the designed structure. Simultaneously, natural ventilation has extended the duration of indoor thermal comfort by 1660 h, significantly enhancing the indoor thermal comfort conditions.

The study investigates the indoor air exchange rates during the transitional seasons due to their optimal natural ventilation potential. Table 2 illustrates the air exchange rates within the building during these transitional seasons under natural ventilation conditions. As depicted in the figures, the area within the building experiencing air exchange rates exceeding 2 h$^{-1}$ amounts to 12,124.48 m$^2$, constituting 63.77% of the total building area of 19,012.41 m$^2$. Hence, optimized designs of buildings can achieve air exchange rates surpassing two times per hour under natural ventilation conditions.

**Table 2.** Table of the number of air changes under typical working conditions in summer.

| Area Ratio of Air Changes Greater Than 2 per Hour | | |
|---|---|---|
| Area with More than 2 Air Changes per Hour (m$^2$) | Total Area (m$^2$) | Area Ratio (%) |
| 12,124.48 | 19,012.41 | 63.77 |

## 6. Conclusions

Natural ventilation shows high application potential in public buildings because of its highly efficient indoor environment quality enhancement and large building energy-saving expectations. How to determine the land use and space planning with building natural ventilation considerations for energy saving is an important research area from the architectural perspective in the early building design stages. In this paper, taking a practical new construction project of a science museum, located in Guangzhou, China, as a typical illustrative example, the detailed architecture and structure design is conducted and optimized with natural ventilation considerations under local climatic conditions. Based on the modeling and simulation, the preliminary results show that the atrium-centered architectural design is favorable for natural ventilation, in terms of the building main opening having an orientation consistent with local dominant wind directions. Therefore, this typical case provides a design approach for large public buildings in the early design stages to enhance indoor thermal comfort and reduce building energy consumption. Specifically, during the initial design phase of large public buildings, incorporating the design of venti-

lation corridors and aligning the main openings of the building with the prevailing local wind direction emerges as an effective design strategy. Such optimization design could facilitate an average air exchange rate over $2\,\text{h}^{-1}$ driven by city wind pressure. Moreover, through building performance simulation and comparison, the natural ventilation in this case building can contribute to about 41.2% air conditioning energy saving ratio due to the free cooling effect. Simultaneously, natural ventilation significantly improved the indoor thermal comfort.

This study presents a typical case in which, during the initial design phase of public building construction, the cooling effects and energy-saving potential of natural ventilation are considered from an architectural perspective. The case aims to elucidate design methodologies and strategies for incorporating natural ventilation factors in the early stages of public building design. However, in the practical operation of buildings, the effectiveness of natural ventilation for cooling and its energy-saving potential are influenced by the external climatic conditions of the building site. Therefore, applying the specific design details introduced in this paper to cooling effects and energy-saving potential achieved under different climatic conditions may result in variations.

Nevertheless, the planning concepts and design methods proposed in this paper, which involve incorporating ventilation corridors in the initial design of public buildings to enhance the cooling effects of natural ventilation and increase energy-saving potential, are universally applicable. This implies that similar design strategies may achieve comparable results even under diverse climatic conditions. Thus, this study provides a universal design reference and application paradigm for projects considering the cooling effects and energy-saving potential of natural ventilation in the early stages of public building design.

**Author Contributions:** Conceptualization, M.Z. and Y.Z.; methodology, W.H.; software, Y.H. and J.X.; validation, Y.H., M.Z. and Y.Z.; formal analysis, W.H.; investigation, J.X.; resources, W.H.; data curation, Y.Z.; writing—original draft preparation, M.Z. and W.H.; writing—review and editing, J.X. and Y.Z.; visualization, Y.H.; supervision, Y.Z. All authors have read and agreed to the published version of the manuscript.

**Funding:** This research was funded by the National Natural Science Foundation of China (no. 52108032) and the National Social Science Foundation of China (no. 23BGL283).

**Institutional Review Board Statement:** Not applicable.

**Informed Consent Statement:** Not applicable.

**Data Availability Statement:** Climatic data are available on the Guangzhou Meteorological Information Community Service Network website (http://www.tqyb.com.cn/sqfw/climate/index.html, accessed on 26 October 2023).

**Conflicts of Interest:** The authors declare no conflicts of interest.

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
