# Peer review of "Natural Ventilation for Cooling Energy Saving: Typical Case of Public Building Design Optimization in Guangzhou, China"

_applsci, doi:10.3390/app14020610_

Round 1

Reviewer 1 Report (Previous Reviewer 2)

Comments and Suggestions for Authors

This revised manuscript has been improved but still authors are unable to signify need of this study as addition of new knowledge. The paper is still not suitable for publication in present for but may be considered after changes.

I have attached manuscript with some minor mistakes using Red Highlighter. Proofread paper again for other mistakes. 

Here are some major comments:

Abstract

The authors presented, “The preliminary results show that such an atrium-centered architectural design could facilitate average air exchange rate over 3 h-1 via natural ventilation effect”, as their main conclusion both in Abstract and Conclusion but results to support this claim are not presented in Results of Discussion Section.

Introduction

The introduction section still lack background to the problem and need of this study. The need of study must be highlighted.

Methodology

Section 2.1 and Section 2.3 are very long and contain irrelevant information. The methodology should contain scientific methods not just description. The description of CFD analysis is not presented well.

Results

The paper is about energy saving but the Results Section does not show any comparison of how much energy is saved using optimization. This energy saving should be calculated and presented in Results or Discussion Section.

Conclusion

The main result presented in the conclusion Section are not discussed in Results Section. Please refer to the comment about Abstract.

Comments on the Quality of English Language

The punctuation and spelling mistakes need to be corrected. 

Author Response

The author sincerely appreciates all the valuable suggestions for further improving our manuscript. We have carefully addressed each comment and made significant revisions accordingly. Please refer to the attached document for a detailed point-by-point response and the marked-up revised manuscript.

Let us know, and Happy New Year!

Reviewer 2 Report (New Reviewer)

Comments and Suggestions for Authors

The research explains an analysis/ study of a public building in Guangzhou, China and its architectural approach considering a strategic need to solve natural ventilation to increase the quality of the indoor ambiance while optimizing and saving energy. Design strategies from preliminary architectural sketches should be a must while designing.

I consider the topic interesting and relevant, but less original, considering the high number of studies regarding this subject. The originality of the paper lies mainly in the analysis of the case study. I don’t own information from other articles that may have made similar analysis before.

Further studies should be considered in the future, with more applicability and use of the knowledge obtained while analysing all the data referring to the specific subject. This I think that is very important for the main topic of the article. I suggest that the authors should also include personal projects and ventilation studies, even made on smaller scale projects.

The conclusions seem consistent and the reference list seems appropriate.

Hoping that I was helpful, I wish the authors a lot of inspiration for improving and publishing the article.

Author Response

The authors sincerely appreciate the positive comment on our present study. Considering your improvement suggestion, along with the other two reviewers' advice, we have spared no time and efforts to carefully address each comment and made significant revisions accordingly. Please see the attached revised manuscript in marked mode. You are also invited to have access to the last other review reports and our response at this review round. Many thanks again for all your time and efforts regarding our submission.

Let us know and happy new year.

Reviewer 3 Report (New Reviewer)

Comments and Suggestions for Authors

This paper reflects a broad overview of both controlled and uncontrolled ventilation in buildings. It seems a little long, which could be referenced and reduced in length.

The research is focused on a specific case. The definition of the specific case should be improved in order to make the research replicable.

On the other hand, the building under study has been designed with various bioclimatic techniques. It is not defined what role these bioclimatic techniques play in the subsequent energy simulations. The modelling of these bioclimatic strategies is highly complex. In case they have not been taken into account in the simulation, they should be clearly stated.

The methodology is not clear in the sense that all the steps that are then carried out, especially after the simulation with Fluent, are not clearly laid out. Simulations are carried out that are not presented in this section, which are even the main results of the research.

As for the CFD simulation, it would be very interesting, since it is an atrium, to obtain results in section, in order to really study the behaviour of the atrium. There is also some doubt as to how an urban environment would influence the results of the CFD simulation.

It is not explained how the results obtained by CFD are treated in the subsequent simulation carried out in CTM methodology. In this simulation, the specific simulation parameters and the software used should also be presented. The operating schedule for the natural ventilation systems, infiltrations, etc. should also be shown.

As for the results, there are some inconsistencies that should be clarified, for example between figure 10, where it can be clearly seen that the situation of discomfort increases in winter, and figure 11, where the energy consumption for heating hardly varies.

Author Response

The authors sincerely appreciate all the rewarding suggestions towards our manuscript further improvement. We have carefully addressed each comment and made significant revisions accordingly. Please see the attached file for the detailed point-by-point responses, along with the marked revised manuscript. 

Let us know and happy new year.

Round 2

Reviewer 1 Report (Previous Reviewer 2)

Comments and Suggestions for Authors

This revised manuscript has been improved and is suitable for publication in the present.

Reviewer 3 Report (New Reviewer)

Comments and Suggestions for Authors

In my opinion, the result of the article has improved, especially in its form, although I think it could be further improved in terms of the geometrical definition of the building to make the research replicable, i.e. another group of researchers could introduce the case in the same calculation software and achieve the same results.

On the other hand, the new graph that has been generated greatly improves the understanding of the paper.

This manuscript is a resubmission of an earlier submission. The following is a list of the peer review reports and author responses from that submission.

Round 1

Reviewer 1 Report

Comments and Suggestions for Authors

The subject covered in the article is of great interest to the AEC industry, in particular to designers. It is intended to motivate and validate the use of natural ventilation in large public buildings to reduce energy consumption for indoor conditioning. In specific, it presents a case study that profits from a central atrium to promote wind movement upward, generating a necessary cooling effect in a rather hot climate context.

However, the author should greatly improve the manuscript, especially to give the right value to the work performed. In the current state, the manuscript is ineffective in every section. For instance, the introduction is incomplete, it does not provide sufficient context, motivation and objective of the work done and presented (why public buildings, how have others validated or highlighted their results).  The methodology was structured and written as a complementary section to the introduction, making it more a background section. More indications on the tools utilized, the settings for them, the assumptions made, and others would have been appreciated (e.g., including clear diagrams, plans with dimensions). The results are presented rapidly and for a single, non-representative, wind condition which lowers the quality and validity of the results and conclusions presented. 

I believe the authors, should also consider the following comments if they wish to improve their work:

1. figure and table captions are not self-explanatory. Also, some of their numbering is wrong (e.g., fig 6b).

2. Most images are of low quality.

3. In line 75, authors mention land use and space planning but what do you mean and how are you implying to relate it to natural ventilation? Do you mean due to wind exposure? exterior surface roughness?

4. You mention this work as a typical illustrative example, but how can it be generalized as guidelines for further use? It feels very project specific.

5. Sections 2.1 and 2.2 are better of as background, not methodology.

6. In geographical information, neighboring cities are mentioned but feel irrelevant as data. Instead, no information of the building surrounding context is given (highly relevant for wind flow).

7. Not a clear methodology is presented on a way of showing the evaluation of the case study, with before and after intervention results or metrics.

8. Present local and international references for climate zones or categories (e.g., Koppen Geiger).

9. Some phrases are redundant (e.g., line 261 – “… (H is the building height)…”)

10. Fig. 4 must have dimensions in it.

11. Lines 282 to 284 present model settings, but without a reason behind.

12. Why the only test done is using wind coming from the southeast when northeast and southwest are the most relevant?

13. The CTM was never presented in the methodology.

14. it is weird that with natural ventilation, winter temperatures are higher. It would be nice to see ventilation rates as well. This should be further discussed or explained.

15. Energy modelling must be more documented.

16. Why is there a difference in lighting and domestic hot water energy use?

17. Paragraph from lines 434 to 444 is identical to 445 to 456.

18. It is stated that an optimization was done, but none is shown in the work.

19. No clear info on how the air exchange rate was calculated.

20. Confusing If the study case is meant to be specific or general.

Comments on the Quality of English Language

English language is clear and the text is well written. However, some phrases should be revised and also punctuation. E.g., revise lines 195-198.

Reviewer 2 Report

Comments and Suggestions for Authors

This manuscript looks like a report and lacks in novelty or addition of new knowledge. This is a case study for a region of China which can be published but the method should be clear which is not a case for this manuscript. This paper is not suitable for publication in its present form but may be considered any major changes.

Here are some major comments:

Introduction

The introduction section lack of background to the problem and need of this study. The novelty of is missing which is not very important for case studies but at least need of study must be highlighted.

Methodology

Section 2.1 and Section 2.3 are very long and contain irrelevant information. The methodology should contain scientific methods not just description. The description of CFD analysis is not presented well.

Conclusion

The main result presented in the conclusion Section are not discussed in Results Section.

Comments on the Quality of English Language

The quality of English of this manuscript is appropriate and easy to understand. 

Reviewer 3 Report

Comments and Suggestions for Authors

the paper introduces a design reference and application prototype for new construction with natural ventilation considerations, that aims to reduce energy consumption of HVAC systems.

the paper focuses on  Guangzhou in China with certain climate parameters

Figure 4 is supposed to show the CFD simulation, however it is not clear and need both clarification and explanation

authors mentioned something about Fig 4c (where is that?)

Fig 5 is not clear at all and in the caption it is written: flow chat.... (is that a typo mistake?)

the explanation provided for Figure 11shows 28% cooling energy used in the optimized design building, i believe clear calculation should be provided to clarify the reduction from 41% to 28%

Comments on the Quality of English Language

Some sentences must be rephrased, specially in the conclusion par